# Deep Learning Based Infrared Thermal Image Analysis of Complex Pavement Defect Conditions Considering Seasonal Effect

**DOI:** 10.3390/s22239365

**Published:** 2022-12-01

**Authors:** Sindhu Chandra, Khaled AlMansoor, Cheng Chen, Yunfan Shi, Hyungjoon Seo

**Affiliations:** 1Faculty of Engineering and Design, University of Bath, Architecture and Civil Engineering, Bath BA2 7AY, UK; 2Department of Civil Engineering and Industrial Design, University of Liverpool, Liverpool L69 3BX, UK; 3Department of Civil Engineering, Xi’an Jiaotong Liverpool University, Suzhou 215000, China

**Keywords:** summer pavement defect detection, winter pavement defect detection, machine learning, thermal analysis, multichannel image fusion

## Abstract

Deep learning techniques underpinned by extensive data sources encompassing complex pavement features have proven effective in early pavement damage detection. With pavement features exhibiting temperature variation, inexpensive infra-red imaging technology in combination with deep learning techniques can detect pavement damages effectively. Previous experiments based on pavement data captured during summer sunny conditions when subjected to SA-ResNet deep learning architecture technique demonstrated 96.47% prediction accuracy. This paper has extended the same deep learning approach to a different dataset comprised of images captured during winter sunny conditions to compare the prediction accuracy, sensitivity and recall score with summer conditions. The results suggest that irrespective of the prevalent weather season, the proposed deep learning algorithm categorises pavement features around 92% accurately (95.18% in summer and 91.67% in winter conditions), suggesting the beneficial replacement of one image type with other. The data captured in sunny conditions during summer and winter show prediction accuracies of DC = 96.47% > MSX = 95.24% > IR-T = 93.83% and DC = 94.14% > MSX = 90.69% > IR-T = 90.173%, respectively. DC images demonstrated a sensitivity of 96.47% and 94.20% for summer and winter conditions, respectively, to demonstrate that reliable categorisation is possible with deep learning techniques irrespective of the weather season. However, summer conditions showing better overall prediction accuracy than winter conditions suggests that inexpensive IR-T imaging cameras with medium resolution levels can still be an economical solution, unlike expensive alternate options, but their usage has to be limited to summer sunny conditions.

## 1. Introduction

Like any engineering structure, during their service life, pavements are prone to damage and deterioration through repetitive usage; if unattended, this could lead to defect propagation needing extensive reparation (or even become unrepairable damage) leading to various safety incidents, traffic accidents, etc. Maintaining pavements in safe operable condition is essential for any national infrastructure development. With technological convergence (combining other engineering technologies with civil engineering), over the past two decades, various SMART detection (and monitoring) solutions have taken precedence in automatically detecting pavement damages from image (still images/videos). Machine learning technology uses a series of images as training data to allow the model (result of the algorithm) to learn about features within the images and predict the category of a new image when it conforms with pre-learned patterns. Extending the capability of these algorithms further, a deep learning approach assesses an image in multiple layers that run deeper and have interconnectivity between layers that aid in better learning. Attempting with basic images captured for automatic pavement damage detection, Chen et al. [1] demonstrated one such combination that initially converts the image into grayscale, followed by contrast enhancement through histogram equalization, then specific road line filtration and a final mean filtering to remove pavement shape change. Intensity-threshold-based algorithms (ITAs), edge-detection-based algorithms (EDAs) and region growing algorithms (RGAs) are some of the complex image segmentation algorithms that are evidenced to have better detection speed, reliability and repeatability [2]. The edge-detection-based algorithm (EDA) also identifies brightness changing points in the image to form curved line segments (edges) using mathematical methods like gradient, Laplacian, etc., through operators, namely Sobel, Prewitt and Canny [3]. Finally, region growing algorithm (RGA) image processing is a cluster algorithm where initial seed points are selected and decision logic is included to compare and add the neighbouring pixels to grow the cluster region [4]. Thus, segmentation quality is solely dependent on the reasonable selection of the initial seeds and the high possibility of over-segmentation/under-segmentation due to a division of clustering results [4,5]. Despite being limited to surface damages, automatic damage detection through image processing/segmentation benefits from quicker objective detection than manual methods, with significant emphasis on continuous improvement on algorithm efficiency and technological exploitation for better results. Consequently, tailoring algorithms has not only complicated pavement image batch processing but also made it less adaptable for all image types [6]. Extending these techniques to pavement damages, by employing deep learning, detection/categorization accuracy up to 99% can be attained [7]. Deep convolutional neural networks (DCNN) are one of the deep learning methods wherein the complexity of the data can be captured through higher levels of feature abstraction [8]. Deep learning algorithms have been gainfully incorporated in civil engineering; for instance, Gopalakrishnan et al. [9] employed a VGG16 network to automatically detect pavement distresses, Cha et al. [8] employed DCNN architecture for detecting concrete cracks in compromised lighting conditions, Zhang et al. [10] employed pre-trained AlexNet for classifying cracks (included sealed cracks), Majidifard et al. [11] employed YOLO (you only look once) and U-net as two-step networks to categorise, and Chen et. al. [12,13] employed a fusion network with acceleration using wavelet transform and VGG16 to identify and detect transverse cracks and manholes. Unanimously, these employed methods transfer the learning sought from pre-trained images to classify or identify elements within the new scenarios. However, complex factors such as oil markings, joints, manholes, etc., warrant a combination approach like that of Zhu et al. [14] that combines DCNN and laser-scanned range images for crack identification by avoiding oil stains/shadows. Guan et al. [15] combined using automatic pavement detection based on stereo vision 3D imaging and deep learning for pothole volume measurements. Although not for pavement, with emphasis on the level of convolutions, [16] employed a 22-layer-deep network to classify and detect features with higher mean average precision (mAP). Alternatively, although time-consuming, initially regularizing deep CNN for a two-stage training method presented better classification results [17]. Recently, studies to measure cracks through thermal infrared images have been conducted [18,19]. Research is being conducted to measure the damage of structures through a 3D point cloud [20,21,22,23]. Point cloud data is also automatically analysed through machine learning algorithms [24].

Additionally, pavement damage has been found to exhibit different temperature profiles [15], i.e., internal temperature of pavement damage varying from that of the pavement surface temperature. Contrastingly, non-damages (such as road marks, oil stains, etc.) may not show any temperature variation, thus signifying that temperature difference can be treated as an attribute for categorisation. As temperature variation can be captured using a relatively inexpensive infrared thermal (IR-T) imaging camera, pavement non-damages can be differentiated from damages without applying any filtering methods. Furthermore, using high-resolution images for deep learning could result in uncompromised accuracy levels. Chen et al. [25] employed commonly used IR-T (and digital) cameras with medium resolution levels in summer sunny conditions to attain high damage detection accuracy of 98.34%; by using the dataset after augmentation, the detection model seems to be more stable to achieve 98.35% precision, 98.34% recall and 98.34% F1-score. This demonstrates that by presenting more learning opportunities for the algorithm through a significantly higher number of input images, (dataset being larger), better accuracy can be achieved. Thus, by performing many tests, it was demonstrated that the results of higher accuracy based on the CNN algorithm can successfully detect (automatically) nine categories of the pavement features (Chen et al. [25,26]). Applying the same deep-learning algorithm (originally written to analyse images captured during summer conditions, Chen et al. [25]) on the images captured during winter sunny conditions, this paper compares the accuracy and precision levels of winter conditions with that of Chen et al. [25]‘s summer sunny conditions. This comparison aims to demonstrate the reliability on use of an inexpensive infra-red (and digital) camera irrespective of the prevailing weather season.

The main contributions of this paper include:This deep learning algorithm categorises pavement features accurately irrespective of the weather season to illustrate the feasibility of replacing one image type with other beneficially.Sunny conditions during summer and winter presented prediction accuracy for DC images, followed by MSX and then IR-T.An inexpensive IR-T imaging camera with medium resolution level can be economical, unlike expensive alternate options; however, its usage is limited to summer sunny conditions.

## 2. Materials and Methods

### 2.1. Overall Procedure

Keeping all parameters similar to those of Chen et al. [25], perform data capturing in winter sunny conditions and execute the data through the same algorithm to obtain accuracy levels for comparison. Figure 1 shows the workflow of the proposed work.

Each pavement damage was captured as an RGB image and thermal image (registered by the device at the acquisition stage). A 50% transparent thermal image was then superimposed on the RGB image to form a fused image, thus forming three distinct data sets for comparison experiments. Additionally, nine different categories of pavement features (damages and non-damages) were considered and 500 images were captured per category. The image data was then put through data augmentation to increase the data sample size. The data set was divided in the ratio of 6:2:2 representing the training set, test set and validation set. The EfficientNet-based learning mechanism was then used to train the model using a tenfold cross-validation training method (making the trained model more stable). The optimal model was then selected for model validation. This workflow is identical to the one proposed in Chen et al. [25].

### 2.2. Experimental Setup and Data Collection

Similar to Chen et al. [25], all pavement data were acquired in Liverpool, UK (marked as red asterisks in Figure 2).

Nine pavement features were considered, namely (1) transverse cracks, (2) longitudinal cracks, (3) alligator cracks, (4) joints or patches, (5) potholes, (6) manholes, (7) shadows, (8) road markings and (9) oil stains. The FLIR ONE camera connected to a cell phone was used for data capture, wherein FLIR ONE’s in-built application displays real-time thermal infrared images and saves both RGB and thermal images in one image by default. The images were then extracted through the MATLAB API interface provided by FLIR. Since this experiment uses a passive heat source for the acquisition, the temperature data acquired is dependent on the collection time. In this case, the data was collected between the months of December 2021 and March 2021 when the ambient temperature was in the range of 2 to 13 degrees Celsius. A total of nine categories of pavement features were acquired, with 500 RGB images for each category that were then extracted for infra-red (IR) and multi-spectral dynamic (MSX) images using software to achieve a total of 13,500.

MSX: The FLIR camera has the capability to add multi spectral dynamic imaging (MSX) that considers both digital camera (DC) image texture and temperature adjustment based on a visualisation ratio to ensure easier target detection without compromising data accuracy. Figure 3 presents the sample image per category by image type.

### 2.3. Data Augmentation, Multi Spectral Dynamic Imaging (MSX) and Test/Training Algorithm

Data Augmentation: The uneven lighting and noise for digital camera (DC) images were corrected using a 2D gamma function that was further combined with the original image [27]. The parameters of the sigma correction function were (15, 80, 150) and (70, 160, 250) and the parameters of Gaussian noise to the original image to simulate noisy images with different signal-to-noise ratio (SNR) values were (20 dB and 35 dB). Additionally, for thermal images, the adjusted colour bar was randomly scaled down by 20% and expanded by 20% to expand the dataset. In this experiment, there were 1500 RGB images and 1500 thermal images in each category after the expansion, totalling 13,500 images. The data classifier was then trained using the original images and the expanded data in the dataset collected in 2.2. Only the collected data were used for evaluation to ensure the accuracy of the model and to avoid the existence of similar images in both train and test sets (the dataset was divided first and then the data was augmented).

Test/Training Algorithm: Further to establishing the 27 categories (3 image types × 9 types of pavement features), with the assistance of the corresponding validation sets of data, algorithm (model) execution commenced through random initialisation that continued onto the training mode (from the training data set), wherein each image was studied for pixel variation, image contrast variation, sharp edge identification, etc., that matured as learning progressed. As evidenced by Jin et al. [28], training algorithms to generate accurate results with conventional RGB images is feasible. (Jin et al. [28] employed an adversarial network model D + GAN for face recognition.)

In simple terms, a neural network works on the logic of assigning numbers to features (or patterns) within the image (during pre-definition) that are digit matched with new images for classification. The features are extracted from the input images in layers (called convoluted layers); i.e., during the first scan of the image (say, layer X1), a number is assigned for each pixel and the number sequence is captured (learning from the first convoluted layer). Further deeper layer scanning of the same image (say, layer X2) captures additional details on the same feature in terms of linked numerical combinations and more importantly spatial locations relative to layer X1 [29]. Hence employing an increased number of deeper layers provides more information about a feature, aiding in better learning. Additionally, these layers are interconnected, enabling effective categorization (into one of the animal datasets) when the data from the new image are passed through these convoluted layers of neural network. This framework of learning pattern is commonly referred to as network architecture.

Based on the understanding of imaging processing detailed in Section 2.3, for this assessment, pixel-level attention will be needed for categorising these images, more importantly both a spatial attention mechanism (pertaining to combining pixel relationships) and channel attention mechanism (pertaining to reliance on specific channels). Zhang et al. [30] devised a methodology to construct a shuffle attention (SA), wherein the input features are placed into distinct groups prior to using a shuffle unit to combine both attention mechanisms (as one block per category) that will be processed through the convolutional neural network approach to finally visualise and validate the built module. Dr. Cheng used this architecture (presented in Figure 1) to develop this adapted algorithm in the Python programming language for this pavement damage categorisation assessment. Zhao et al. [31] demonstrated higher processing speed for every frame analysed using advanced pixel embedment towards clustering interference for analysing medical images as a plug-and-play model. This exhibited increased accuracy with acceptable memory consumption. Additionally, for 3D datasets, Zhao et al. [32] proposed a VoxelEmbed method to achieve performance of 20.6% with 2% containing segmentation annotations, which has been demonstrated to be generalized with efficient memory utilisation.

Residual network (ResNet) architecture is considered one of the effective convolutional neural networks that has been demonstrated to have the ability to assess up to 1202 layers for effective model performance [21]. Notation ResNet-50 refers to ResNet architecture with an inherent ability to learn from 50 layers of image (or computer vision) information. As suggested earlier, the higher the number of layers assessed, the better will be the performance accuracy of the model. As evidenced by Zhang et al. [31], coupling ResNet with a shuttle attention (SA) approach should show better accuracy. An attempt was made to compare these two image processing architectures with this project’s pavement damage/non-damage image test (evaluation) data for 50 epochs. The obtained result was in line with Zhang et al. [31] findings, with SA-ResNet architecture showing better accuracy levels with progressive loss minimisation than the conventional ResNet. Thus, banking on its better performance potential, SA-ResNet architecture was used for this assessment.

Refining the model: As the model (resultant of the algorithm) is trained (or at the commencement of training), modifying pre-set parameters helps in refining them. Following are the key refining parameters (also known as hyperparameters) used for fine tuning the model. The employed architecture of the proposed CNN is presented in Figure 4 and Table 1.
Convolution layers: The number of layers within each image that will be assessed for feature extractions. Fifty layers were employed for this assessment,Loss function type: The convergence of the prediction with ideal output. The cross-entropy loss type was employed for this assessment. This loss function measures the level of deviation between the prediction and the actual image. Specifically, this measures loss as log loss (y-axis) that handles two different probabilistic distributions. Evidently, the higher the log loss, the higher the distinction between the probability distributions.Performing this on the summer conditions data, the profile for all three image types exhibited a converse nature of losses with the accuracy profile and the level of deviation drops (converges) for all three image types. Commencing at a high logarithmic scale of error, the level of convergence reached 0.039, 0.098 and 0.163 for DC, IR-T and MSX image types, respectively, as the model’s learning progressed.Epoch: Refers to the number of times the algorithm puts a test (or evaluation) image through the training data (higher passes results in better results). A hundred epoch passes were employed for this assessment,Batch size: Each epoch pass is executed within an interior loop enabling a batch of on input image processed. A batch size of 48 was employed for this assessment,Optimiser: Application of statistical optimization approach for convergence. A stochastic/paddle type optimiser was employed for this assessment, andMomentum: Parameter that dictates the subsequent step’s direction from the current step (preventing back and forth oscillations), usually a slope measure. A momentum of 0.9 was employed for this assessment.

For the learning phase, at the commencement of execution, the algorithm just scans the top layer of the image. With no convoluted layers applied at this stage, the accuracy levels were low for all three image types. However, when well into the learning phase, as predicted, all image types exhibited decreasing profile for losses and increasing profile for accuracy. However, the level of accuracy to which the model is trained to tends to vary with each image type. In DC images, the pavement features are distinctly visible as pavement showing clear pavement discolouration, darker cracks, blacker shadows, uneven depths at patches, deeper potholes, contrasting road-marks, colour variation with metal-aggregate manholes, etc. These variations show distinct pixel differentiation at the pavement feature’s edges. Hence, the level of accuracy attained by DC images was more than 90%. On the contrary, IR-T images lack distinct/crisp pixel differentiation at the pavement feature’s edges, as the pixel colour difference is based on the pavement surface temperature. Hence, in line with temperature profile, the pixel variation is gradual (not sharp). Thus, the feature extraction from images becomes slightly harder (than DC) for the algorithm, resulting in attainable accuracy levels of under 90%. Finally, benefiting from light clarity enhancement onto the thermal image, MSX images exhibited pixel differentiation at a level that in between DC and IR-T, with accuracy levels closer to 90%.

The loss function, on the other hand, measures the level of deviation between the prediction and actual image. Specifically, cross-entropy loss, adopted for this assessment, measures loss as log loss that handles two different probabilistic distributions. The higher the log loss, the higher the distinction between the probability distributions. The obtained profile for all three image types indicates the converse nature of losses with accuracy profile and that the level of deviation drops (converges) for all three image types. Commencing at a high logarithmic scale of error, the level of convergence reaches 0.039, 0.098 and 0.163 for DC, IR-T and MSX image types, respectively, as the model’s learning progresses. Figure 5 shows the reduction in losses during the training phase for the summer season (green graph) and winter season (purple graph), wherein losses of less than 3 and less than 5 were observed initially, respectively.

The effectiveness of the learned algorithm was further assessed based on its accurate category prediction of the test (evaluation) dataset images. The test (evaluation) dataset consisted of 1701 images (567 DC, 567 IR-T and 567 MSX) that were split when put through the learned algorithm. The algorithm is executed by image type; for example, after being trained using 3933 DC training images, the algorithm will predict the pavement damage/non-damage category of 567 DC test images. Additionally, each of the test (or evaluation) images were put through 100 epochs (number of times each test image is put through the training dataset). Assessing the accuracy profiles of all image types, apart from just one spurious trough in the DC image, the evaluation phase profiles were similar to the ones obtained in the learning phase. In addition, the increasing accuracy profile corroborates the theory that accuracy level generally increases with additional epoch iterations before plateauing. For DC images, although they present a higher accuracy level after 100 epochs, the variation seems high, i.e., relatively unsmooth with wider deviation between the peaks and troughs. Minor differentiations between crack types (for instance alligator cracks with different depths along the damage can easily resemble longitudinal cracks) could be one of the reasons for this.

The loss function, on the other hand, plateaus in early epochs for DC and clarity-enhanced MSX compared with IR-T, where it exhibits a gradual decreasing profile. In addition to not having a sharp temperature variation at pavement damage/non-damage edges, the requirement for needing more spatial attention to categorise them could be the reason for this. Having more epochs for IR-T could potentially smoothen the profile too. Figure 6 shows the reduction in losses during the evaluation phase for the summer season (green graph) and winter season (purple graph). Additionally, Table 2 presents the employed layer information and reasoning speed summary.

## 3. Results

### 3.1. Evaluation Metrics

The employed evaluation metrics incorporating true positive (TP), false positive (FP), true negative (TN) and false negative (FN) values are as follows:(1)$$\mathrm{Accuracy}=\frac{TP+TN}{TP+TN+FP+FN}$$
(2)$$\mathrm{Precision}=\frac{TP}{TP+FP}$$
(3)$$\mathrm{Recall}=\frac{TP}{TP+FN}$$

Having performed data pre-processing on MATLAB 2020a to form the dataset, similar to Chen et al. [8], Python was used for the deep learning operation and the Efficient module was executed using the PyTorch framework on I9 10,850 K CPU and 2080 ti GPU systems. The SGD algorithm was then used to update the network weight with momentum 0.7 and weight decay 1 × 10^−4^ for all our experiments. The initial learning rate was set to 0.01 and adjusted further along during training using a multi-learning rate strategy, where the initial learning rate is multiplied by 0.6 for each iteration (max_iteration was set to 30 k). Furthermore, the parameters of the encoder EfficientNet were initialised using a transfer learning method on ImageNet after pre-training. Additionally, other parameters were initialised randomly. Finally, the batch size of the dataset was set to 8, and to avoid overfitting, the data were scaled up per the steps described in 2.3.

The original and augmented datasets used in this study were randomly divided into training, validation and test sets, accounting for a 60%, 20% and 20% split, respectively. The training and validation datasets were used only for training and installation of the model, while the test set was used to evaluate the predictive performance of samples that were not previously seen by the model. Table 3 presents the original and augmented dataset size (inclusive of RGB, IR and MSX images). With the algorithm being exposed to diverse patterns in the training phase, a clear distinction between training and testing data was essential to avoid overfitting (prediction being clouded when algorithm is already exposed to these images in training phase) and underfitting (inadequate data to predict correctly). Hence, around 80–90% of images captured were used for training and the remainder of images for evaluation/validation.

### 3.2. Comparing of Confusion Matrices for Summer and Winter Season in Sunny Conditions

A confusion matrix is one of the easily comprehensible ways to measure the model performance, wherein the predicted values are compared with the actual values to present the performance in terms of the following four metrics, wherein TP, TN, FP and FN represent correct positive prediction, correct negative prediction, incorrect positive prediction and incorrect negative prediction, respectively. In this assessment, when considering just two variables, namely alligator cracks and other categories (Table 4), the model correctly predicted 61 images as alligator cracks (true positive) and 500 images as other categories (true negative), whereas it incorrectly predicted 4 as alligator cracks (false positive) and 2 as other categories (false negative).

The percentage for correct prediction can be calculated using three main terms, namely precision, sensitivity and overall accuracy. Precision is the measure of correct prediction, i.e., the ratio of true positive to overall positive prediction (61/65 in the mentioned example). Similarly, sensitivity is the measure of positive classes, i.e., the ratio of true positive to overall actuals (61/63 in the mentioned example). Finally, the overall accuracy is the measure of overall correct predictions, i.e., the ratio of true positive and true negative to overall actuals ((61 + 500)/567) in the mentioned example). The same logic can be extended for nine pavement damage/non-damage categories to present in a 9 × 9 matrix in the Figure 7, Figure 8, Figure 9, Figure 10, Figure 11 and Figure 12. The summer sunny condition for DC images (Figure 7) presents an overall accuracy of 96.47%, with 100% prediction rate for well-prominent features such as shadows, manholes and road markings.

However, the winter sunny condition for DC images (Figure 8) presents an overall accuracy of 94.14%, with 100% prediction rate for shadows only.

The overall accuracy level for the summer sunny condition for IR images (Figure 9) was 93.83%, with relatively low levels for joints.

However, with 90.17% overall accuracy for winter sunny conditions for IR-T images (Figure 10), prominent features like shadows and road-marks were predicted with better accuracy levels.

Better than IR-T images, the summer sunny condition for MSX images (Figure 11) shows an overall accuracy of 95.24%, where all pavement features were predicted better than IR-T images.

Finally, the winter sunny condition for MSX images (Figure 12) shows 90.69% accuracy levels.

## 4. Discussion

The data captured in sunny conditions during summer and winter show a prediction accuracy of DC = 96.47% > MSX = 95.24% > IR-T = 93.83% and DC = 94.14% > MSX = 90.69% > IR-T = 90.173%, respectively. This information is compared in Table 5.

On average, the model categorises pavement features around 92% accurately (95.18% in summer and 91.67% in winter conditions) irrespective of the image type, suggesting the potential feasibility of replacing one image type with the other beneficially, irrespective of the weather season. Thus, by employing commonly used cameras with limited resolution, higher accuracy levels can be attained with a larger input dataset. The decrease in accuracy levels from DC to IR-T or MSX is predominantly due to the reduction in pixels that compromises the shape details (mainly edges); when not exposed to higher temperatures (during winter), the clear demarcations disappear.

Irrespective of the weather season, assessing pavement non-damages, prediction sensitivity and prediction precision of road marking and shadows were performed best using the IR-T image dataset, followed by the DC image dataset. This can also be interpreted as the possibility that the capture (through IR camera) of temperature differences between non-damages (such as shadows and road markings) and the pavement surface, resulting in pixel colour distinction, can be used as effectively as DC images (captured through a digital camera) for pavement feature categorisation. Figure 13 presents the sensitivity and precision measures for summer conditions and Figure 14 presents the same for winter conditions.

On the other hand, for other pavement non-damages such as manholes and oil marks, the prediction precision of IR-T images is significantly lower than for DC images. This is due to the lack of significant temperature difference, as oil marks appear as mere pigmentation and manhole material temperature is like pavement surface temperature. Thus, mere IR-T images might not be sufficient for effective categorisation. However, with additional aid of light-enhanced MSX imaging, the categorisation appears to be as effective as DC, with similar precision levels.

With DC images showing better prediction accuracy (reasonably closer variation between summer and winter conditions), reduced prediction accuracy for IR-T and MSX illustrates that image processing is compromised when processing is applied to already-software-processed images. Similar to summer conditions, if the winter conditions data were to be captured on all three image types (DC, IR-T and MSX), they would definitely present better accuracy results. Figure 15, Figure 16 and Figure 17 present the sensitivity and precision for DC, IR-T and MSX images, respectively.

Irrespective of the season and image type, pavement features, namely shadows, exhibited lesser deviations (100% for both seasons for the DC image type especially). Contrastingly, oil marks deviated the most between seasons. Additionally, the orientation of the images being the only difference between pavement features such as longitudinal and transverse can be deemed as the main reason for the lower % accuracy level. In addition, the phenomenon is apparent in both seasons. Furthermore, the summer and winter season seem to observe a similar trend, with the exception of alligator cracks for DC and IR-T image types.

In addition, summertime data benefit from higher visibility and relatively higher inherent pavement temperature to exhibit a vibrant colour palette in both the thermal infrared (IR-T) as well as the multi-spectral (MSX) modes, especially noticeable in the manhole category due to different material schemes in the same frame. Also evident from the results are that with 2.33% variance between weather seasons, the prediction accuracy of DC images was higher than IR-T (3.66%) and MSX (4.55%) image types. It is also possible that IR-T and MSX can be put to better use in wet ground conditions, as they can interpolate the temperature difference and isolate and distinguish the crack features from the moisture when wet. Alternatively, sunny conditions during winter months are more likely to appear dry despite possessing increased moisture content, as the ground could be colder (or even freezing). This will definitely alter both its temperature profile and reflective nature.

Irrespective of the weather season, due to a lack of temperature variation in all pavement non-damages, there was not distinctive pixel variation to categorise more precisely. Lower IR-T prediction precision and prediction sensitivity than DC demonstrates this (refer to Figure 18 and Figure 19). However, the prediction levels of MSX are closer to DC, due to its inherent light enhancement feature.

Assessing pavement damages, the prediction sensitivity and prediction precision of MSX imaging more closely resemble DC images for summer conditions than winter conditions. Despite this, IR-T/MSX cannot be employed on its own, as images of pavement damages require further processing and extracting of additional damage details, which could only be possible with digital imaging technology. However, bigger damages like potholes and longer transverse cracks have been categorised better with IR-T/MSX than DC (irrespective of the weather season).

This assessment suggests that IR-T and IR enhanced MSX imaging effectively categorises pavement non-damages at an accuracy level like DC images irrespective of the weather season. Hence, by employing IR technology, images with non-damages can be filtered out from the image processing dataset before applying any filtering method. Thus, reducing the size of dataset requiring further processing in turn benefits from increased processing speed and overall efficiency. Additionally, deep learning techniques applied to images captured during summer sunny conditions exhibit better and more reliable prediction accuracy of over 95% as opposed to winter sunny conditions. Table 6 presents the detailed comparison of the performance of different input data by class.

## 5. Conclusions

Early detection of pavement damages has been shown to reduce reparation costs and, more importantly, prevent safety incidents that would have happened because of extensive damage propagation over time. With technological convergence, artificial intelligence (AI)-based techniques such as deep learning through computer vision can be gainfully adopted within civil engineering for a faster and more accurate way to automatically detect pavement damages. These deep learning models’ accuracy can be compromised by the presence of inherent unwanted information (noises) such as road-marking, oil stains, shadows, etc. within the images. Additionally, employing high-resolution cameras for better accuracy might not be economical all the time. Thus, to improve the overall efficiency, this project explored the feasibility of (1) filtering out image noises through infrared thermal (IR-T) imaging (based on lack of temperature variation between pavement damages and non-damages) and (2) providing more learning opportunities for the algorithm from a larger input dataset of medium resolution images.

This paper compared 13,500 manually captured images of pavement features (9 × 3 × 500—nine identified categories—three pavement damages and four pavement damages, three image types—DC, IR-T, MSX and 500 images in each category) during summer sunny conditions with a combination of 13,500 manually captured and software-converted images (9 × 500 × 3—4500 manually captured DC images—nine identified categories with 500 images per category, software converted to IR-T and MSX images). These images were captured through a commonly used medium-resolution camera and subjecting them to a convolution neural network—SA ResNet architecture deep learning algorithm to be trained from 87% of the test dataset to effectively predict 13% of images accurately. This project concludes that:The deep learning algorithm categorises pavement features (both damages and non-damages) around 92% accurately (95.18% in summer and 91.67% in winter conditions) irrespective of the weather season to illustrate the feasibility of replacing one image type with the other beneficially. Additionally, despite limited resolution, higher accuracy levels can be attained by providing the algorithm with more learning opportunities through a large input dataset.The data captured in sunny conditions during summer and winter show a prediction accuracy of DC = 96.47% > MSX = 95.24% > IR-T = 93.83% and DC = 94.14% > MSX = 90.69% > IR-T = 90.173%, respectively.From DC image input, the sensitivity was 96.47% for summer conditions and 94.20% for winter conditions. With the capturing method being manual, the deep learning technique can categorise pavement features reliably irrespective of the weather season.From IR-T and MSX image input, over 90% precision (93.95% summer IR-T, 90.23% for winter conditions IR-T, 95.65% for summer MSX and 90.76% for winter MSX) suggests that the prevalence of the temperature profile in pavement features can be gainfully utilised to categorise them irrespective of the weather season.With summer conditions showing better overall prediction accuracy than winter conditions, we suggest that an inexpensive IR-T imaging camera with a medium resolution level can still be economical, unlike expensive alternate options; however, its usage is limited to summer sunny conditions.

Furthermore, provision of a large input image dataset presents more learning opportunities for the deep learning algorithm, resulting in better prediction accuracy, thus aiding in improving the overall effectiveness of the deep learning model.

## Figures and Tables

**Figure 1 sensors-22-09365-f001:**
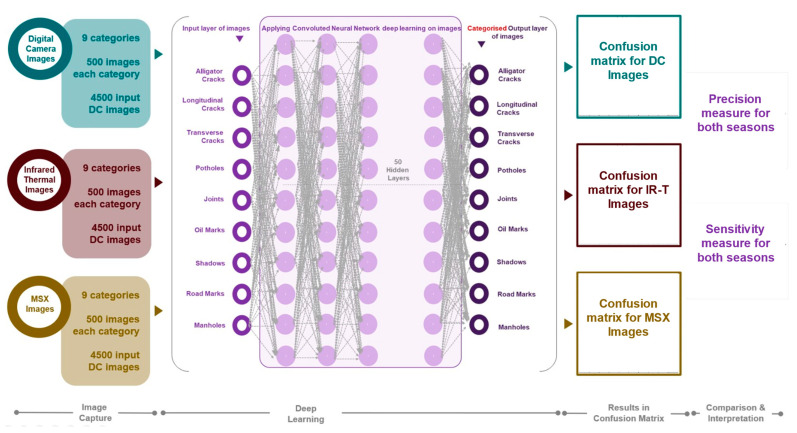
Project workflow.

**Figure 2 sensors-22-09365-f002:**
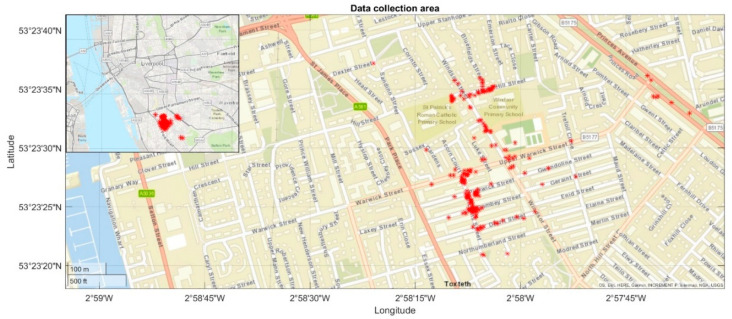
Location of data collection.

**Figure 3 sensors-22-09365-f003:**
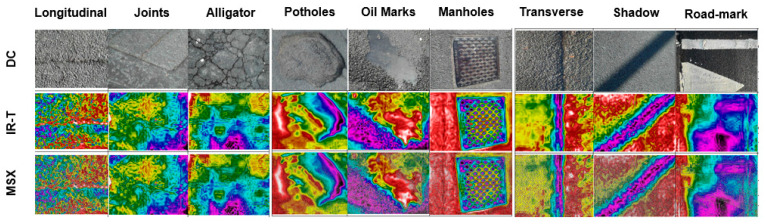
Sample image per category by image type.

**Figure 4 sensors-22-09365-f004:**
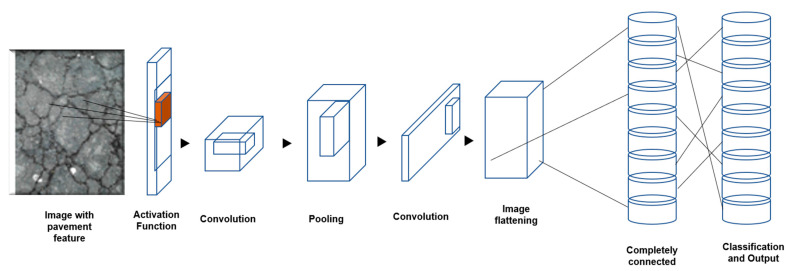
Architecture of the proposed CNN model.

**Figure 5 sensors-22-09365-f005:**
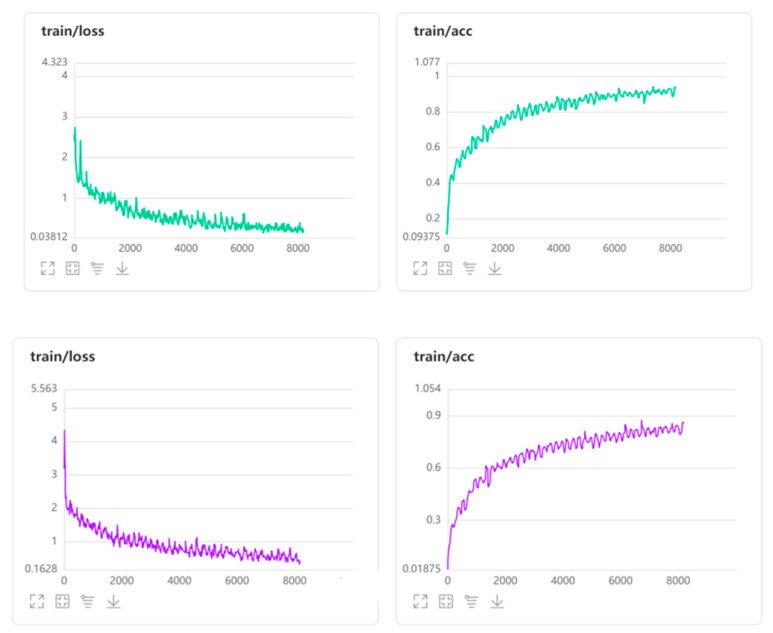
Training phase losses for summer and winter season.

**Figure 6 sensors-22-09365-f006:**
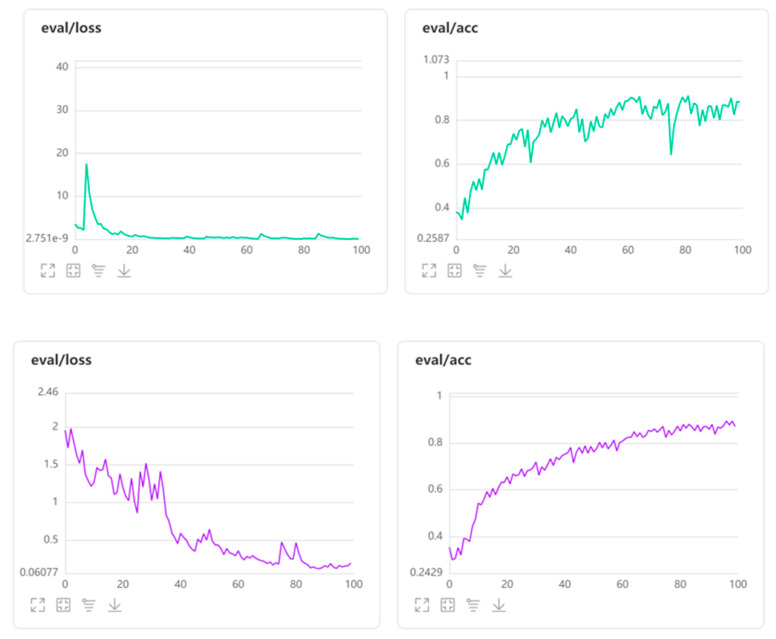
Evaluation phase losses for summer and winter season.

**Figure 7 sensors-22-09365-f007:**
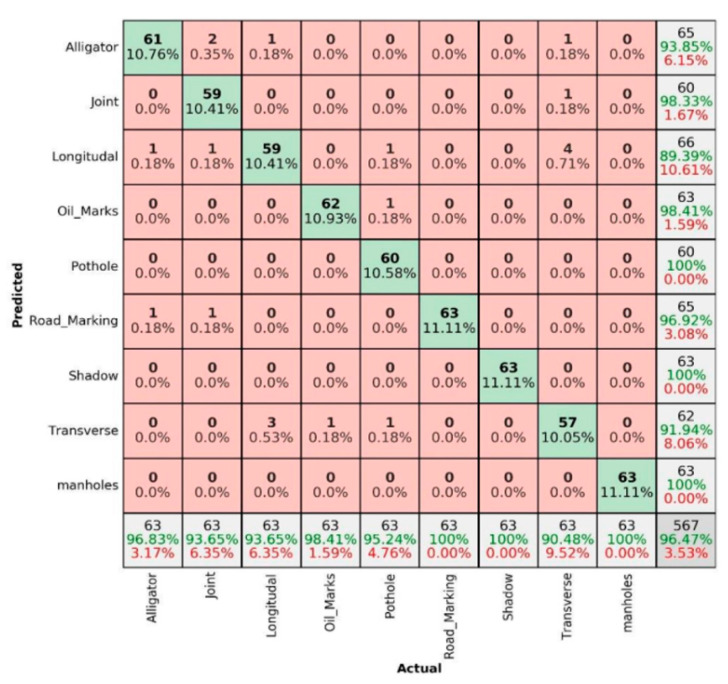
Summer sunny condition—DC images.

**Figure 8 sensors-22-09365-f008:**
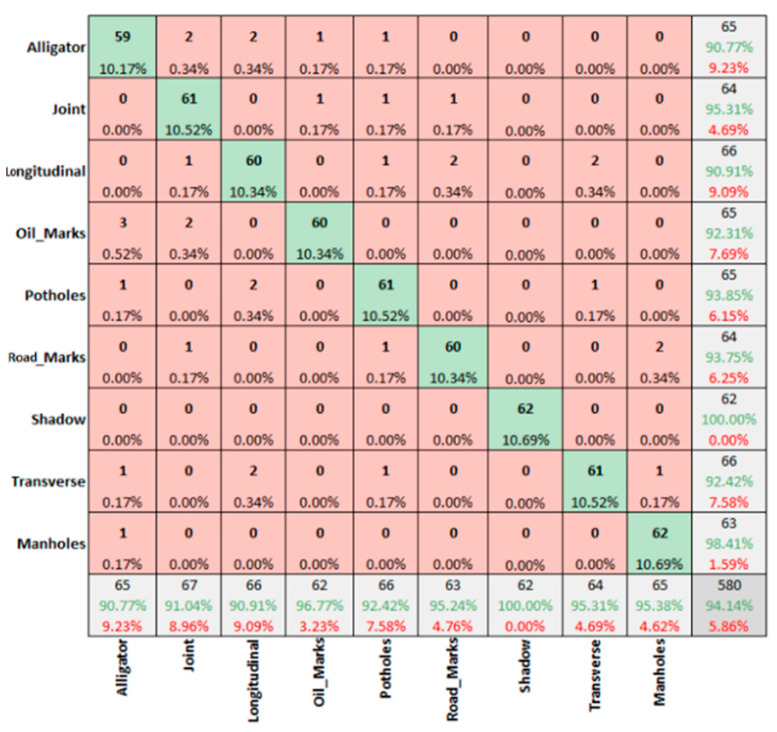
Winter sunny condition—DC images.

**Figure 9 sensors-22-09365-f009:**
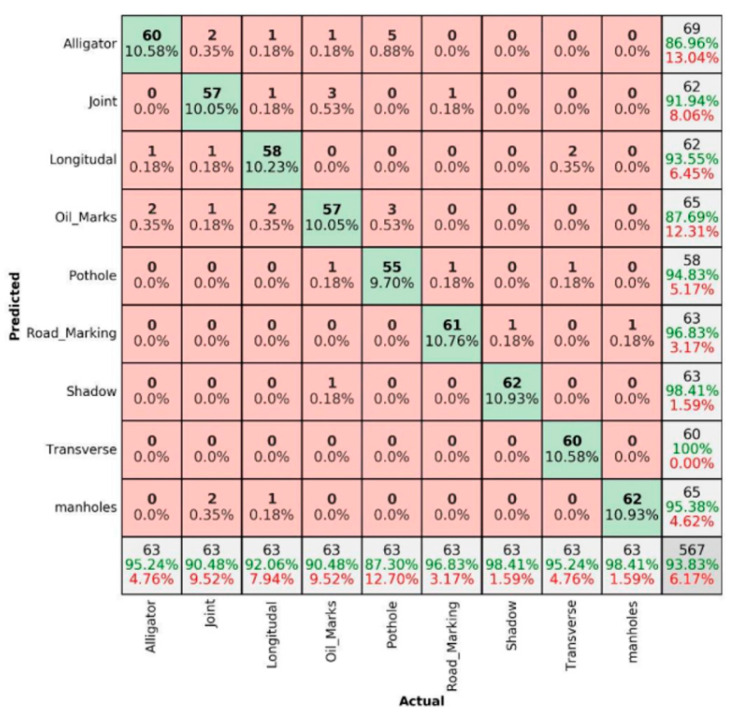
Summer sunny condition—IR-T images.

**Figure 10 sensors-22-09365-f010:**
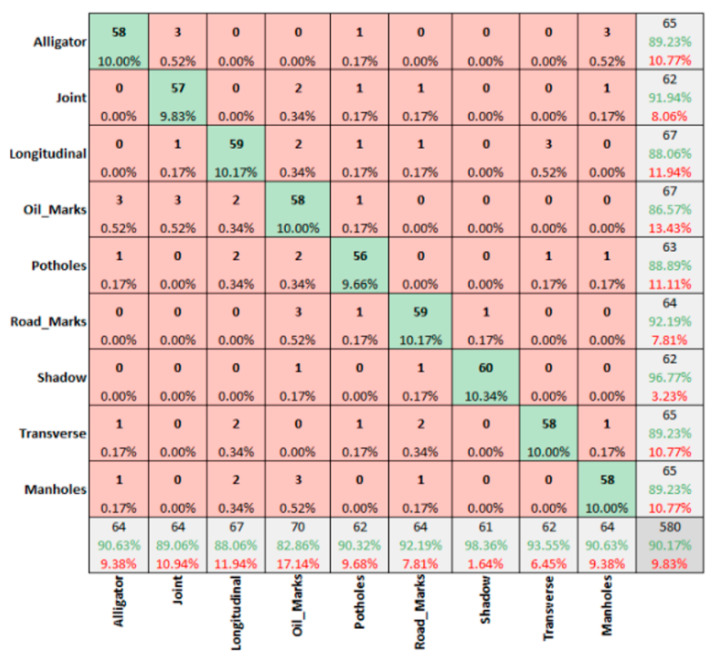
Winter sunny condition—IR-T images.

**Figure 11 sensors-22-09365-f011:**
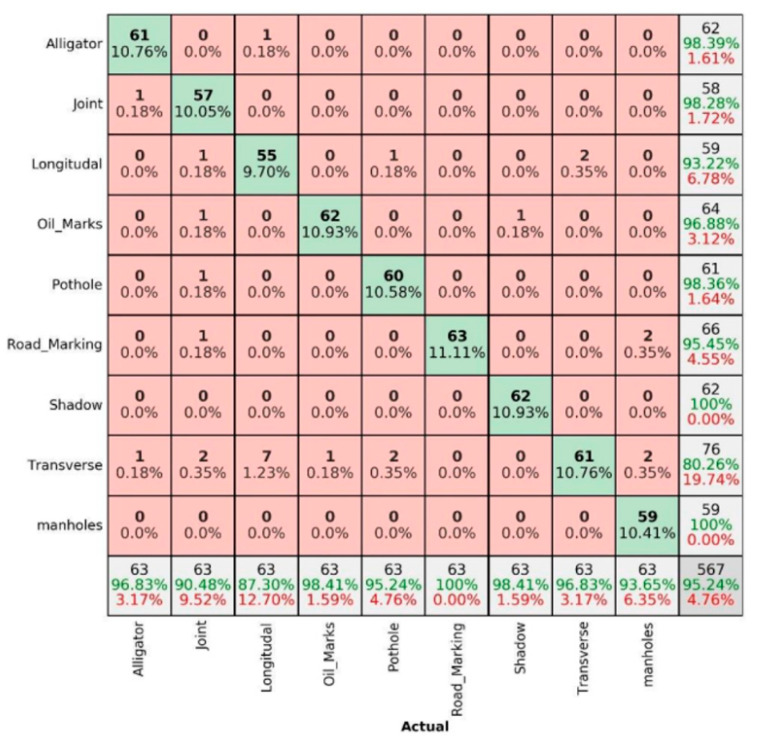
Summer sunny condition—MSX images.

**Figure 12 sensors-22-09365-f012:**
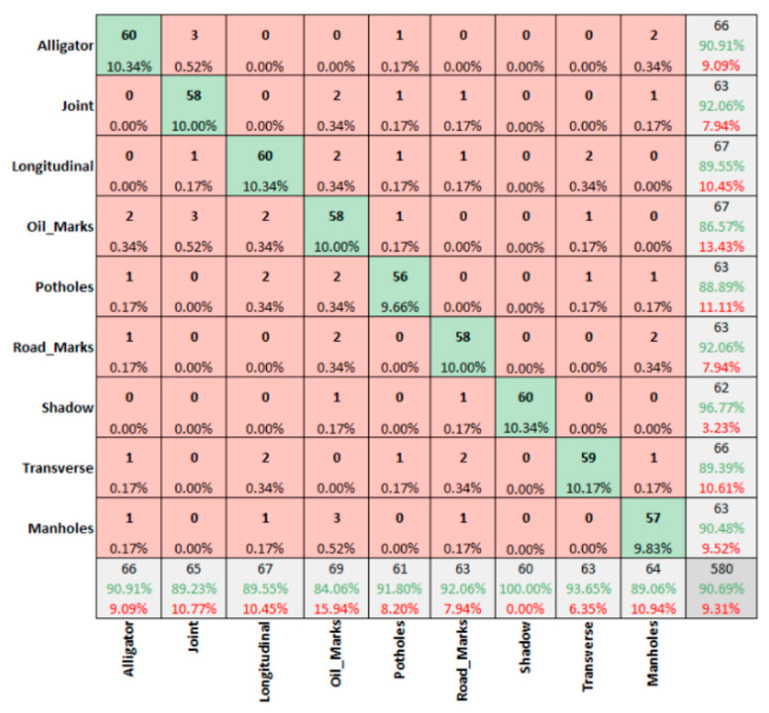
Winter sunny condition—MSX images.

**Figure 13 sensors-22-09365-f013:**
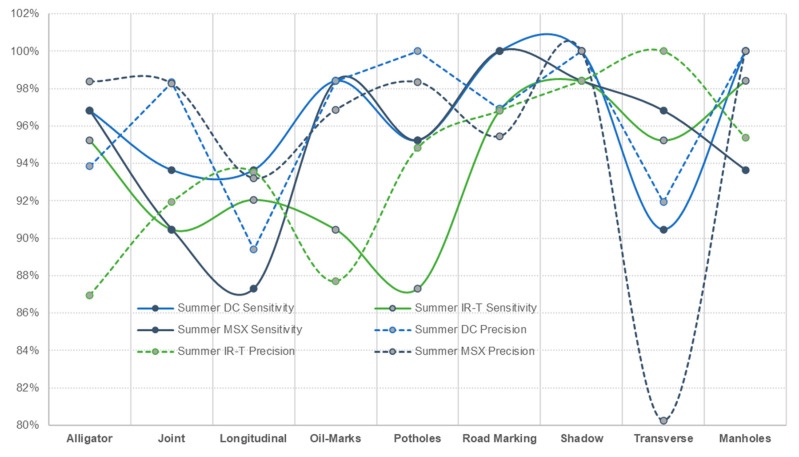
Summer conditions—sensitivity and precision.

**Figure 14 sensors-22-09365-f014:**
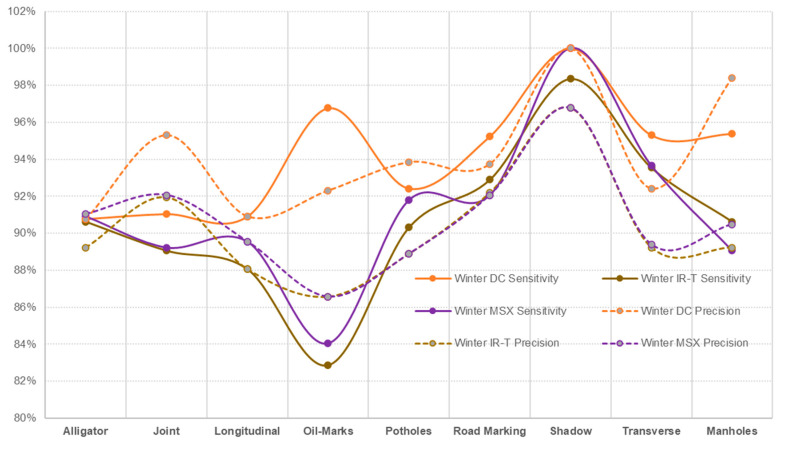
Winter conditions—sensitivity and precision.

**Figure 15 sensors-22-09365-f015:**
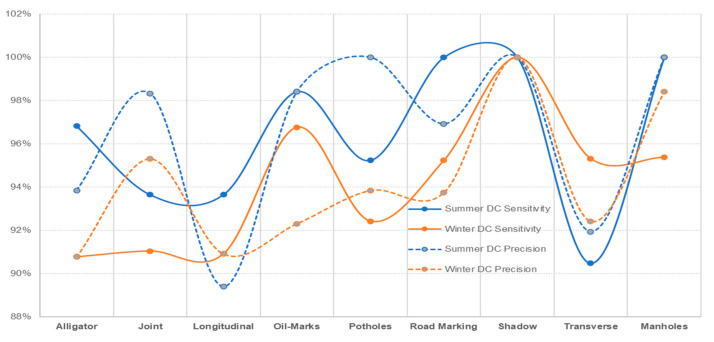
Comparing summer and winter conditions for DC images.

**Figure 16 sensors-22-09365-f016:**
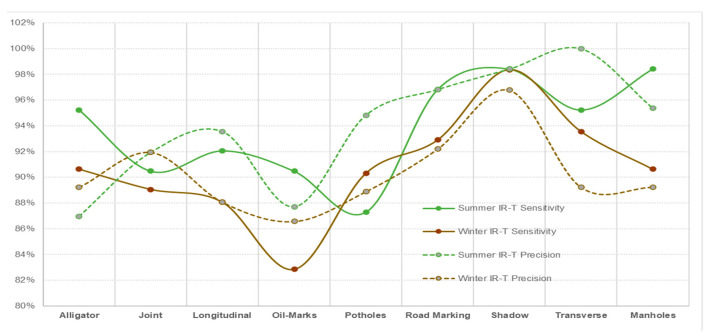
Comparing summer and winter conditions for IR-T images.

**Figure 17 sensors-22-09365-f017:**
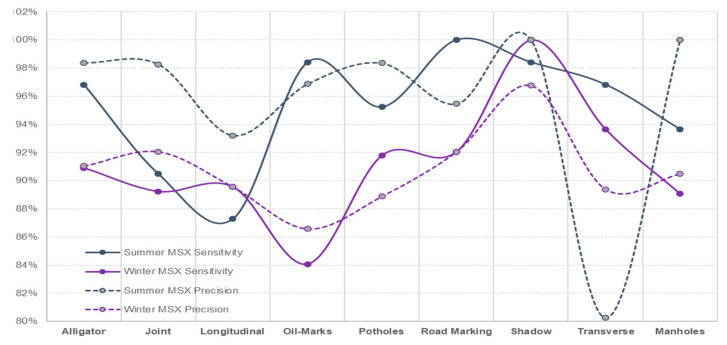
Comparing summer and winter conditions for MSX images.

**Figure 18 sensors-22-09365-f018:**
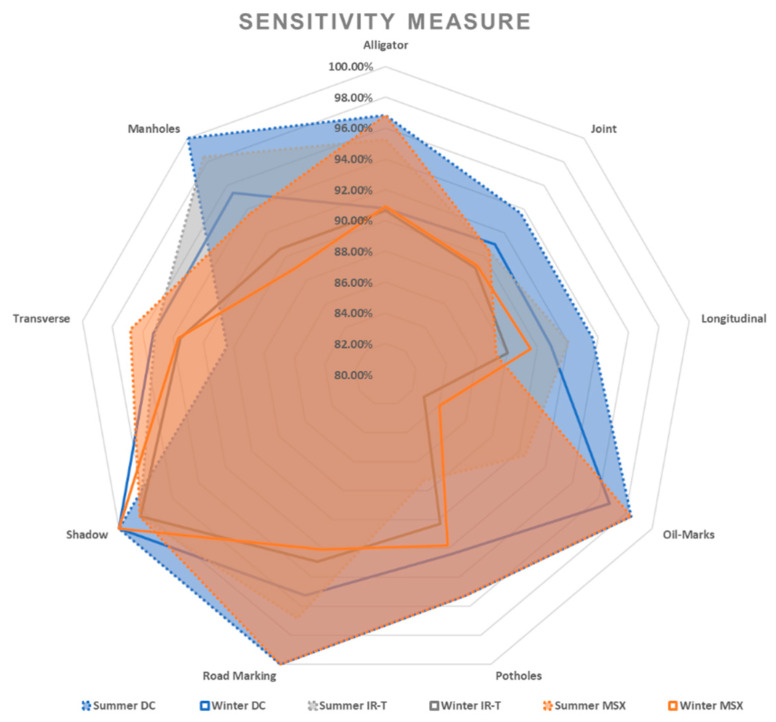
Sensitivity measure.

**Figure 19 sensors-22-09365-f019:**
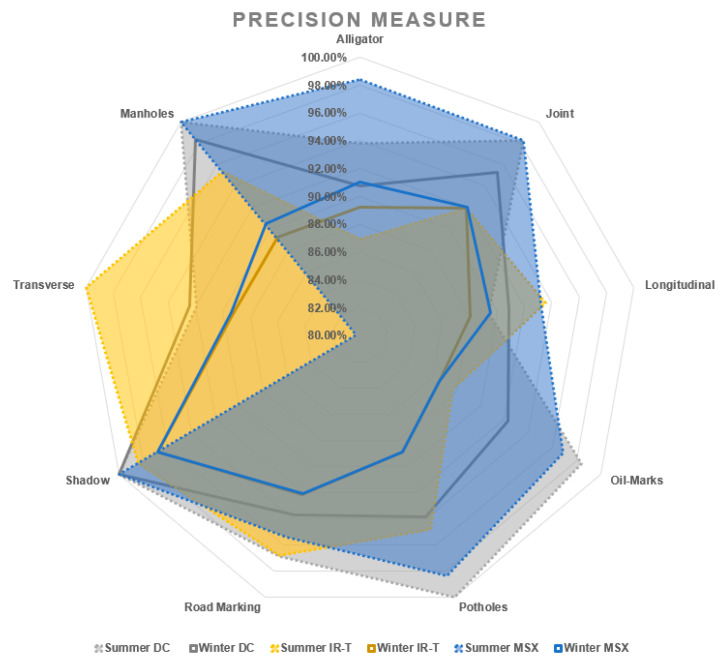
Precision measure.

**Table 1 sensors-22-09365-t001:** Deep learning algorithm—experimental setup and parameters.

Deep Learning Algorithm Parameters	Parameter Details
Dataset considerations	Every 8th image identified as evaluation image
	Evaluation = 567 DC + 567 IR + 567 MSX images
	Training = 3933 DC + 3933 IR + 3933 MSX images
Functional parameter	Cross-entropy loss function
	Stochastic paddle optimiser
Model parameters	Hyperparameter-tuned learning rate
	0.9 momentum
	100 epoch times
	Batch size 48

**Table 2 sensors-22-09365-t002:** Layer information, reasoning speed summary.

Layer (Type)	Output Shape	Parameter
Input layer	[(None, 256, 256, 3)]	0
Functional	(None, 8, 8, 512)	14,714,688
Global average pooling	(None, 512)	0
Dense	(None, 1024)	525,312
Additional dense	(None, 8)	8200
Total parameters: 15,248,200		
Trainable parameters: 533,512		
Non-trainable parameters: 14,714,688		

**Table 3 sensors-22-09365-t003:** Description of the amount of data in the raw dataset and augmented dataset.

	Total	Train (60%)	Validation (20%)	Test (20%)
Original dataset	13,500	8100	2700	2700
Augmented dataset	18,945	11,367	3789	3789

**Table 4 sensors-22-09365-t004:** Confusion matrix—2 variables (DC alligator used for instance).

		Actual (Summer Conditions)			Precision
		Alligator Cracks	Other Categories	Total	
Predicted	Alligator cracks	61	4	65	61/65 = 93.85%
	Other categories	2	500	502	
	Total	63	504	567	
	Sensitivity	61/63 = 96.83%			561/561 = 98.94%

**Table 5 sensors-22-09365-t005:** Comparison of different input data in pavement crack detection.

	Summer Sunny Condition			Winter Sunny Condition		
	Accuracy	Precision	Recall	Accuracy	Precision	Recall
DC	96.57%	96.59%	96.57%	94.14%	92.19%	92.20%
IR-T	93.83%	93.95%	93.83%	90.17%	90.23%	90.63%
MSX	96.83%	96.92%	96.83%	90.69%	90.76%	91.15%

**Table 6 sensors-22-09365-t006:** Comparison of the performance of the different input data for each class.

	Summer Sunny Conditions			Winter Sunny Conditions		
	DC	IR-T	MSX	DC	IR-T	MSX
Alligator	96.83%	95.24%	96.83%	90.77%	90.63%	90.91%
Joint	93.65%	90.48%	90.48%	91.04%	89.06%	89.23%
Longitudinal	93.65%	92.06%	87.30%	90.91%	88.06%	89.55%
Oil marking	98.41%	90.48%	98.41%	96.77%	82.86%	84.06%
Pothole	95.24%	87.30%	95.24%	92.42%	90.32%	91.80%
Road marking	100%	96.83%	100%	95.24%	92.19%	92.06%
Shadow	100%	98.41%	98.41%	100%	98.36%	96.77%
Transverse	90.48%	95.24%	96.93%	95.31%	93.55%	89.39%
Manholes	100%	98.41%	93.65%	95.38%	90.63%	90.48%
Average	96.47%	93.83%	95.24%	94.20%	90.17%	90.76%

## Data Availability

Not applicable.

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
