# Peer review of "Deep Learning Based Infrared Thermal Image Analysis of Complex Pavement Defect Conditions Considering Seasonal Effect"

_sensors, 2022, doi:10.3390/s22239365_

Round 1
Reviewer 1 Report
The following comments must be carefully revised. (1) The following work should be cited and discussed, including “Improvement of generalization ability of deep CNN via implicit regularization in two-stage training process,” IEEE Access, vol. 6, pp. 15844-15869, 2018. “Faster Mean-shift: GPU-accelerated clustering for cosine embedding-based cell segmentation and tracking.” Medical Image Analysis 71 (2021): 102048. “VoxelEmbed: 3D instance segmentation and tracking with voxel embedding based deep learning,” International Workshop on Machine Learning in Medical Imaging. Springer, Cham, 2021. “Pseudo RGB-face recognition,” IEEE Sensors Journal, 2022, doi: 10.1109/JSEN.2022.3197235. (2) The specific model of deep learning (e.g. the number of convolution kernels, step size, etc) and the reason for designing this architecture should be further introduced. (3) The optimization curve of the objective function of the deep learning model with the iterative process should be presented. (4) The experimental setup of the proposed deep learning model and comparison methods should be reported in detail, such as batch size. (5) The structural complexity and reasoning speed of the model are encouraged to display. (6) More technical details should be added in the second part to help understand the method used.
Author Response
Thanks for the comments and the detailed response to the reviewer's comments is attached.

Reviewer 2 Report
The paper is a very interesting and meaningful work. But the paper still needs to consider the following questions.
1. It is necessary to further elaborate the research status and add more recent references.
2.The comparison of experiments should increase the comparison of more advanced methods.
Author Response

(The authors gave the same response as above.)

Reviewer 3 Report
In this paper the authors present a Deep learning based infrared thermal image analysis of complex pavement defect condition, for the winter and summer seasons, the results showed a better generalization for the samples taken in sunny summer. here some comments:
· the text is clearly written which facilitates its understanding, however, a minor spelling revision to the text is recommended
· Authors are recommended to increase the number of articles referenced in the introduction
· Authors are encouraged to highlight the main contributions of the paper in the introduction.
· in figure 1 there are parts that are not clear, for example the results of the confusion matrix, it is recommended to improve its quality
· It is recommended to discuss in greater depth the results obtained,
Author Response

(The authors gave the same response as above.)

Round 2
Reviewer 1 Report
The authors should carefully review the comments and make careful modifications. In the current version, many key comments were only answered perfunctorily without corresponding modifications in the manuscript, including comment#1(all the recommended related work should be cited in the paper), #3(The figure of the optimization curve is still not added), #5(the reasoning speeds should be summarized in a table), and #6(technical detail of proposed method should be added).
Author Response
Thanks for the comments and a file for detailed responses is attached.
